# Training-free Graph Anomaly Detection: A Simple Approach via Singular Value Decomposition

## ABSTRACT

Graph anomaly detection has been widely applied in real-world applications, where deep learning-based methods have demonstrated promise. However, prior methods often suffer from various limitations, such as poor detection accuracy, long training time, complicated training schemes, and lack of scalability. To combat this dilemma, we propose TFGAD, a simple yet effective training-free approach for graph anomaly detection. Particularly, TFGAD comprises two transformation matrices, each of which serves to process one type of node feature (attributes or local structure). Notably, these matrices can be optimally determined via singular value decomposition, thus requiring no prior training. Further, we tailor a lightweight anomaly scoring function, which integrates the reconstruction error of attributes with the projection length of local structures to quantify graph anomalies. Extensive experiments demonstrate that TFGAD leads to significant improvements over state-of-the-art reconstruction-/contrastive-based deep learning baselines while reaching much less runtime and memory overhead.

## CCS CONCEPTS

• **Security and privacy** → **Web application security**; • **Computing methodologies** → **Anomaly detection**.

## KEYWORDS

anomaly detection, attributed graphs, training-free, singular value decomposition

### ACM Reference Format:

Anonymous Author(s). 2018. Training-free Graph Anomaly Detection: A Simple Approach via Singular Value Decomposition. In *Proceedings of Make sure to enter the correct conference title from your rights confirmation emai (Conference acronym 'XX)*. ACM, New York, NY, USA, 11 pages. https://doi.org/XXXXXXX.XXXXXXX

## 1 INTRODUCTION

Graph anomaly detection recently has received increasing attention due to its wide applications in various security-related fields [31, 39, 52]. Notable examples include social spam detection [23], financial fraud detection [50], and network intrusion detection [4, 16]. The goal of graph anomaly detection is to discern abnormal nodes that significantly deviate from the majority of nodes in a graph. Typically, there are two main types of abnormal nodes [27, 29],

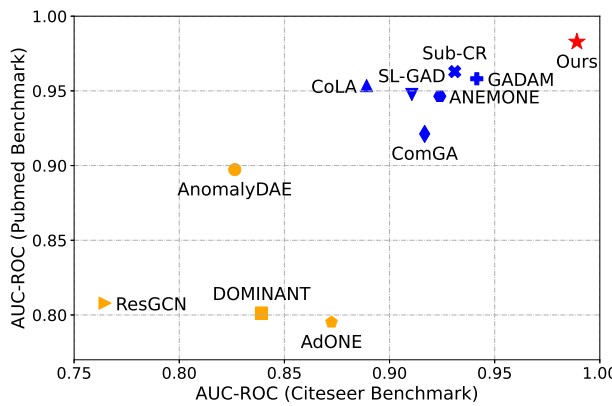

**Figure 1: AUC-ROC performance of our method and representative strong baselines on two popular benchmark datasets.**

contextual and structural anomalies. The former refers to nodes whose attributes differ significantly from their neighbors, while the latter relates to densely connected nodes contrasting with sparsely connected regular nodes.

With the booming of deep learning techniques, learning-based graph anomaly detection has recently dominated the research focus. Current methods can be roughly categorized into reconstruction- and contrastive-based [37, 59], where the latter attracts more attention due to its better detection capabilities. The representative approach CoLA [29] works by leveraging contrastive learning to model the relationship between nodes and their local structures. Recent methods [9, 10, 25] make further improvements by incorporating more powerful contrastive strategies. **Despite their empirical success, these methods require long training time and complicated training/inference strategies, leading to poor efficiency and flexibility in practice.** In contrast, reconstruction-based methods can naturally avoid the above issues [5, 37], which have demonstrated promise with a simple and straightforward pipeline. These methods build upon the idea from residual analysis [44], where anomalies manifest as large residual/reconstruction errors compared to normal counterparts. The pioneering work DOMINANT [7] identifies abnormal nodes by reconstructing both their attributes and local graph structures via graph neural networks (GNNs). Subsequent methods [6, 13, 17] then enhance the model robustness to anomalies for improved accuracy. **However, these methods often underperform contrastive-based methods and lack scalability due to the prohibitive cost of reconstructing large-scale graphs.**

The above dilemma faced by the two mainstream deep learning-based anomaly detection methods prompts a question: **Are key**

properties such as accuracy, efficiency, scalability, and simplicity inherently compatible for graph anomaly detection? Targeting this problem, we propose TFGAD, a simple yet effective training-free approach for graph anomaly detection. Not only does TFGAD outperform previous works (Figure 1), but it is also simpler, more efficient, and scalable, requiring no training parameters. Motivated by the simplicity and potential of reconstruction-based methods in detecting graph anomalies, we begin by investigating key issues in these methods and their negative impact on performance. The results demonstrate that a minimalistic, GNN-free, and modality-separate framework can be superior for graph anomaly detection, which separately encodes/reconstructs attribute and graph structure data with minimum required transformations. Based on this, TFGAD comprises two transformation matrices, each of which serves to process one type of node feature (attributes or local structure). Remarkably, these matrices can be optimally determined via singular value decomposition (SVD). Thus, it removes the need for deep learning techniques but can still achieve superior detection performance. As a further improvement, we employ randomized SVD [15] to increase computation efficiency on large-scale graphs. Besides, a lightweight scoring function is also adopted to improve detection accuracy and efficiency. It substitutes the reconstruction process of local structures by projecting them into a low-dimensional subspace, providing diverse advantages for graph anomaly detection. We summarize our contributions as follows:

- We re-examine the issues of existing reconstruction-based graph anomaly detection methods and their negative impact on performance, which highlights the superiority of a minimalistic, GNN-free, and modality-separate framework for graph anomaly detection.
- We propose a training-free approach for graph anomaly detection, TFGAD, which is simple, efficient, scalable, and easy to implement. Remarkably, TFGAD requires no training parameters and can be fully developed with SVD. It also contributes a novel lightweight scoring function for better detecting graph anomalies.
- We conduct extensive experiments on various benchmark datasets, including two large-scale datasets (ogbn-Arxiv and ogbn-Products) with millions of edges. The results show that TFGAD reaches state-of-the-art performance with much less runtime and memory overhead than baselines. Specifically, TFGAD demonstrates improvements in AUC-ROC ranging from 4.5% to 35.1% and achieves speedups of 3.0× to 68.0× across various benchmarks, all without requiring GPU overhead.

## 2 RELATED WORK

### 2.1 Graph Anomaly Detection

Graph anomaly detection aims to identify nodes that deviate from the majority ones. Early methods employ shallow techniques, e.g., ego-network analysis [35], matrix factorization [28], residual analysis [24], and CUR decomposition [34], which struggle to handle complex graph information. With the rapid development of deep learning techniques in the field of data mining [8, 36, 57], deep learning-based approaches have been widely applied in graph anomaly detection. A typical framework takes reconstruction as the learning objective, where the pioneering work DOMINANT [7] employs a graph autoencoder to reconstruct both attribute and graph structure data for anomaly detection. AnomalyDAE [13] extends this by incorporating a graph attention mechanism [49] for encoding complex graph structure information. AdONE [1] employs two autoencoders to separately process attributes and graph structure, followed by a random walking strategy to enhance graph structure information. Recent methods [6, 13, 17, 40] further improve detection accuracy by employing more robust frameworks against overfitting to anomalies. The success of contrastive-based anomaly detection in computer vision and other related domains sheds light on the potential of contrastive learning for graph anomaly detection [20, 38, 45]. CoLA [29], the representative approach, works by capturing normal patterns between nodes and their neighboring substructures via contrastive learning. ANEMONE [21] extends this by introducing multi-scale contrasts for improved focus on node-level information. SL-GAD [59] incorporates the reconstruction process of attribute data into CoLA's framework, aiming to leverage the strengths of both reconstruction- and contrastive-based methods for improved accuracy. Subsequent methods [9, 10, 25] make further improvements based on CoLA by employing more powerful contrastive learning frameworks. The above methods, typically leveraging GNNs, are prone to obscure anomalous information due to message passing. Recently, methods particularly designed to overcome the shortcomings of message passing have received increasing attention for graph anomaly detection. A notable example is GADAM [5], which identifies local anomalies via inconsistency mining and applies adaptive message passing to capture global anomaly signals. Although the above methods achieve reasonable performance, they all have limitations, such as inaccuracy, inefficiency, and lack of scalability [5, 27, 29]. It is still worthwhile to explore an approach with various compelling advantages for graph anomaly detection.

### 2.2 SVD-based Graph Processing

SVD, a widely-used matrix decomposition technique, plays a crucial role in various data analysis tasks [18, 56, 58]. With the growing research interest in graph-structured data, SVD recently has garnered renewed attention for its simplicity and effectiveness [12, 14, 33, 53], where a typical SVD-based application is for recommendation systems, which employ bipartite graphs to characterize the interactions between users and items. The representative method [2] builds a lightweight recommender system by leveraging SVD to process incomplete data streams online. The subsequent method [60] improves this by incorporating more efficient incremental techniques. More recently, with the rapid development of deep learning techniques, SVD has been widely combined with GNNs to enhance recommender systems. For instance, LightGCL [3] utilizes SVD as an effective data augmentation scheme to enrich the user-item interaction information for better representation learning via GNNs; SVD-GCN [33] reveals the connection between GNNs and SVD for recommendation and replaces the core design of GNNs with a flexible truncated SVD for improved simplicity and efficiency. When it comes to security-related domains, SVD has proven effective in improving the robustness to graph anomalies. The pioneering

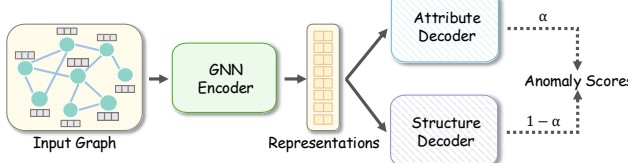

**Figure 2: General framework of reconstruction-based anomaly detection methods.**

work [43] exploits SVD and low-rank approximation of the user interaction matrix to spot suspicious behavior; the recent method [12] leverages SVD to defend against adversarial attacks on graphs by focusing on their top singular components. However, despite the broad applications of SVD in processing graph-structured data, its potential for detecting graph anomalies remains underexplored.

## 3 PRELIMINARIES

### 3.1 Problem Formulation

We first introduce the main notations used in this paper. We use plain, bold lowercase, and bold uppercase letters to denote scalars, vectors, and matrices, respectively, e.g., $k$ and $\eta$ are scalars, $\boldsymbol{x}$ and $\boldsymbol{z}$ are vectors, $\boldsymbol{X}$ and $\boldsymbol{W}$ are matrices. $\|\boldsymbol{x}\|_2$ is the Euclidean norm of vector $\boldsymbol{x}$, where $\|\boldsymbol{x}\|_2 = \sqrt{\sum_i |\boldsymbol{x}_i|^2}$ and $\boldsymbol{x}_i$ is the $i$-th entry of $\boldsymbol{x}$. $\|\boldsymbol{X}\|_F$ is the Frobenius norm of matrix $\boldsymbol{X}$, where $\|\boldsymbol{X}\|_F = \sqrt{\sum_{i,j} |\boldsymbol{X}_{ij}|^2}$ and $\boldsymbol{X}_{ij}$ is the $(i, j)$-th entry of $\boldsymbol{X}$.

With the aforementioned notations, let $\mathcal{G} = \{\mathcal{V}, \boldsymbol{X}, \boldsymbol{A}\}$ be an attributed graph. $\mathcal{V} = \{v_1, v_2, \ldots, v_n\}$ is a set of $n$ nodes. $\boldsymbol{X} = [\boldsymbol{x}_1, \boldsymbol{x}_2, \ldots, \boldsymbol{x}_n]^{\mathrm{T}} \in \mathbb{R}^{n \times d}$ is an attribute matrix, where its $i$-th row indicates the attributes of node $v_i$, characterized by vector $\boldsymbol{x}_i \in \mathbb{R}^d$. $\boldsymbol{A} \in \mathbb{R}^{n \times n}$ is an adjacency matrix, where $\boldsymbol{A}_{ij} = 1$ if there is an edge between nodes $v_i$ and $v_j$, and $\boldsymbol{A}_{ij} = 0$ otherwise. The $i$-th row of $\boldsymbol{A}$ indicates the local structure of node $v_i$, characterized by vector $\boldsymbol{a}_i \in \mathbb{R}^n$, and $\boldsymbol{A} = [\boldsymbol{a}_1, \boldsymbol{a}_2, \ldots, \boldsymbol{a}_n]^{\mathrm{T}}$. By training on $\mathcal{G}$ (if required), the goal of graph anomaly detection is to build a scoring function $\tau(\cdot) : \mathbb{R}^d \mapsto \mathbb{R}$ to quantitatively measure abnormal degrees of nodes in $\mathcal{V}$. Notably, no ground-truth information is accessible during training, thus a fully unsupervised approach is required.

### 3.2 Reconstruction-based Graph Anomaly Detection

Reconstruction-based graph anomaly detection methods discern abnormal nodes using reconstruction errors of their attributes and local structures. Figure 2 illustrates the general framework of these methods, which follows a simple and straightforward pipeline: a GNN-based encoder learns latent representations of nodes, while two decoders separately reconstruct original node attributes and local structures from these representations. The objective function of this framework is formulated as:

$$\mathcal{L}_{rec} = (1 - \alpha)\|\boldsymbol{X} - \hat{\boldsymbol{X}}\|_F^2 + \alpha\|\boldsymbol{A} - \hat{\boldsymbol{A}}\|_F^2, \tag{1}$$

where $\hat{\boldsymbol{X}}$ and $\hat{\boldsymbol{A}}$ represent reconstructed attributes and local structures, and $\alpha > 0$ is a hyper-parameter balancing the weights of different terms. For anomaly detection, the reconstruction error of

a test node $v_i$ serves as its anomaly score:

$$\tau(v_i) = (1 - \alpha)\|\boldsymbol{x}_i - \hat{\boldsymbol{x}}_i\|_2^2 + \alpha\|\boldsymbol{a}_i - \hat{\boldsymbol{a}}_i\|_2^2, \tag{2}$$

where a higher score indicates a greater likelihood of $v_i$ being abnormal.

### 3.3 Singular Value Decomposition

Singular value decomposition (SVD) is a popular matrix decomposition technique with broad applications in data mining [18, 56, 58]. Let $\boldsymbol{W} \in \mathbb{R}^{p \times q}$ be a real-valued matrix of rank $r$. The SVD of $\boldsymbol{W}$ is given by:

$$\boldsymbol{W} = \boldsymbol{U}\boldsymbol{\Sigma}\boldsymbol{V}^{\mathrm{T}} = \sum_{i=1}^{r} \sigma_i \boldsymbol{u}_i \boldsymbol{v}_i^{\mathrm{T}}, \tag{3}$$

where $\boldsymbol{U} \in \mathbb{R}^{p \times r}$ is an orthogonal matrix containing the left singular vectors of $\boldsymbol{W}$ on its columns $\{\boldsymbol{u}_i\}_{i=1}^{r}$. The diagonal matrix $\boldsymbol{\Sigma} = \mathrm{diag}(\sigma_1, \ldots, \sigma_r)$ contains the singular values ($\sigma_1 \geq \sigma_2 \geq \ldots \geq \sigma_r > 0$). $\boldsymbol{V} \in \mathbb{R}^{q \times r}$ is another orthogonal matrix containing the right singular vectors of $\boldsymbol{X}$ on its columns $\{\boldsymbol{v}_i\}_{i=1}^{r}$.

According to the Eckart-Young theorem [11], the best rank $k$ approximation to $\boldsymbol{W}$ is given by the truncated SVD:

$$\boldsymbol{W}_k = \boldsymbol{U}_k \boldsymbol{\Sigma}_k \boldsymbol{V}_k^{\mathrm{T}} = \sum_{i=1}^{k} \sigma_i \boldsymbol{u}_i \boldsymbol{v}_i^{\mathrm{T}}, \tag{4}$$

where $\boldsymbol{U}_k$ and $\boldsymbol{V}_k$ contain the top $k$ left and right singular vectors of $\boldsymbol{W}$, respectively. For any rank $k$ matrix $\boldsymbol{B}$, the inequality $\|\boldsymbol{W} - \boldsymbol{W}_k\|_F \leq \|\boldsymbol{W} - \boldsymbol{B}\|_F$ holds.

## 4 METHODOLOGY

In this section, we start with a detailed investigation of issues in reconstruction-based graph anomaly detection in Section 4.1. To combat these issues, we propose TFGAD, a simple and effective training-free approach for detecting graph anomalies. We detail TFGAD's framework in Section 4.2 and how TFGAD discerns anomalies and its compelling advantages in Section 4.3. Finally, we provide a complexity analysis of TFGAD in Section 4.4.

### 4.1 Investigating Issues in Reconstruction-based Graph Anomaly Detection

Reconstruction-based graph anomaly detection identifies abnormal nodes by measuring their reconstruction errors. However, these errors are often inseparable between normal and abnormal nodes, leading to poor detection performance. To explore how this inseparability arises, we investigate key issues in existing methods and their negative impact on performance:

- **Issue 1: Learning from Abnormal Neighbors**: Existing methods typically adopt GNNs to learn node representations by aggregating information from neighboring nodes. When anomalies exist, representations of normal nodes can be distorted via their potential abnormal neighbors, thus impairing detection accuracy.
- **Issue 2: Reconstructing from Entangled Representations**: Existing methods employ GNN-induced representations for reconstruction, which are considered to fuse information from both attribute and graph structure data.

**Table 1: AUC-ROC performance of DOMINANT and its competing variants across three popular benchmark datasets. The results are averages over five runs. The best result per dataset is boldfaced, while the second-best is underlined.**

| Method | Cora | Citeseer | Pubmed |
|---|---|---|---|
| LINEAR-DOMINANT | 0.9404 | 0.8535 | 0.8688 |
| MINLIN-DOMINANT | **0.9572** | 0.8805 | 0.8696 |
| SEPARATE-DOMINANT | 0.9567 | **0.9473** | **0.9133** |
| DOMINANT | 0.8493 | 0.8391 | 0.8013 |

However, this fusion can obscure information from different data types, making it difficult to capture their inherent patterns for effective anomaly detection.

- **Issue 3: Overfitting to Anomalies**: Current methods often employ overly complex architectures, leading to the overfitting problem. As a result, abnormal nodes can also be reconstructed well, with reconstruction errors similar to those of normal nodes.

To investigate how these issues impact reconstruction-based graph anomaly detection, we conduct an ablation study on the pioneering and representative method DOMINANT [7]. We propose three variants of DOMINANT: **(1) LINEAR-DOMINANT**: This variant substitutes all GNN layers in DOMINANT with linear layers, making it possible to access the contribution of GNNs to anomaly detection. In this case, the GNN encoder of DOMINANT is replaced by a linear encoder that learns node representations solely from attributes. **(2) MINLIN-DOMINANT**: This variant simplifies LINEAR-DOMINANT by minimizing the number of its layers, which helps to validate the effectiveness of reducing the complexity of model architectures in mitigating overfitting to anomalies. **(3) SEPARATE-DOMINANT**: It extends MINLIN-DOMINANT by additionally introducing a single linear layer to learn the representation of node local structures, while the corresponding structure decoder is also implemented as a single linear layer. This allows for the separate reconstruction of attribute and graph structure data from different representations (modality-separate), which can validate its effectiveness in detecting graph anomalies.

Table 1 illustrates the AUC-ROC performance of DOMINANT and its competing variants across three popular benchmark datasets: Cora, Citeseer, and Pubmed. All variants significantly outperform DOMINANT, e.g., with AUC-ROC improvements of 10.73%-12.70% and 12.07%-14.07% on Cora and Citeseer, respectively. Notably, SEPARATE-DOMINANT demonstrates superior performance on Citeseer and Pumbed, with respective AUC-ROC improvements of 12.89% (from 0.8391 to 0.9473) and 13.98% (from 0.8013 to 0.9133) compared to DOMINANT. On Cora, SEPARATE-DOMINANT also performs competitively, with a minimal margin of less than 0.0005 in AUC-ROC compared to the best result of MINLIN-DOMINANT. These results demonstrate that those above-discussed issues can substantially impair the performance of reconstruction-based graph anomaly detection. Moreover, the promising results of the proposed variants suggest that GNN-free, minimalistic, and modality-separate frameworks can be superior for detecting graph anomalies.

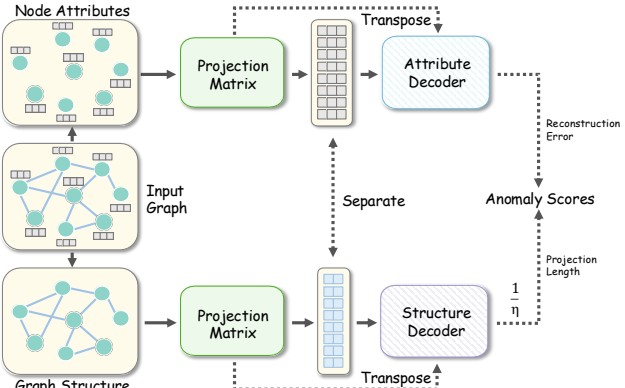

**Figure 3: Overall framework of TFGAD.**

## 4.2 Reconstruction via Truncated SVD

Based on the findings in Section 4.1, we propose TFGAD, which is minimalistic, GNN-free, and capable of separately processing (encoding and reconstructing) node attributes and local structures to detect graph anomalies. Figure 3 illustrates the overall framework of TFGAD. Basically, it includes two transformation matrices optimized with the objective of minimizing the reconstruction error over node attributes and local structures:

$$\min_{W_{\mathcal{A}}, W_{\mathcal{S}}} \|X - XW_{\mathcal{A}}W_{\mathcal{A}}^{\mathrm{T}}\|_2^2 + \|A - AW_{\mathcal{S}}W_{\mathcal{S}}^{\mathrm{T}}\|_2^2, \quad (5)$$

where $W_{\mathcal{A}} \in \mathbb{R}^{d \times k_{\mathcal{A}}}$ and $W_{\mathcal{S}} \in \mathbb{R}^{d \times k_{\mathcal{S}}}$ represents two linear transformations which project the node attributes $\{x_i\}_{i=1}^n$ and local structures $\{a_i\}_{i=1}^n$ into $k$- and $q$-dimensional subspaces, respectively. Their transposes are then applied to map the projected data back to their original spaces for reconstruction. According to the Eckart-Young theorem [11], this objective has a closed-form analytic solution via truncated SVD. Here, the optimal $W_{\mathcal{A}}^*$ and $W_{\mathcal{S}}^*$ are simply the top $k$ and $q$ right singular vectors of the attribute matrix $X = [x_1, x_2, \ldots, x_n]^{\mathrm{T}}$ and adjacency matrix $A = [a_1, a_2, \ldots, a_n]^{\mathrm{T}}$, respectively.

However, performing (exact) SVD on large matrices is computationally expensive. To address this, we adopt randomized SVD [15], which approximates the input matrix with a smaller one before performing SVD, thereby significantly reducing computational overhead. Formally, the randomized SVD of matrix $W$ is given by:

$$\tilde{U}_k, \tilde{\Sigma}_k, \tilde{V}_k = \text{ApproxSVD}(W, k), \quad (6)$$

where $k$ is the required rank for the decomposed matrices, and $\tilde{U}_k$, $\tilde{\Sigma}_k$, and $\tilde{V}_k$ are the approximated versions of $U_k$, $\Sigma_k$, and $V_k$. This provides an approximately optimal solution for objective (5), i.e., $\{\tilde{V}_{\mathcal{A}} \in \mathbb{R}^{d \times k_{\mathcal{A}}}, \tilde{V}_{\mathcal{S}} \in \mathbb{R}^{d \times k_{\mathcal{S}}}\}$.

## 4.3 Anomaly Detection

We now describe the procedure of performing graph anomaly detection leveraging the approximately optimal weight matrices $\tilde{V}_{\mathcal{A}}$ and $\tilde{V}_{\mathcal{S}}$. In particular, given a test node $v_i$ and its corresponding attribute and local structure vectors $x_i$ and $a_i$, we compute the

anomaly score $\tau(v_i)$ according to:

$$\tau(v_i) = \|x_i - \tilde{V}_{\mathcal{A}}\tilde{V}_{\mathcal{A}}^{\mathrm{T}}x_i\|_2^2 + \frac{1}{\eta}\|\tilde{V}_{\mathcal{S}}^{\mathrm{T}}a_i\|_2^2, \qquad (7)$$

where $\eta > 0$ is a balancing hyper-parameter. Note that reconstructing the local structure vector $a_i$ can be computationally expensive due to its high dimensionality, which equals the total number of nodes in the input graph and can grow excessively as the graph scales. To address this, we adopt the projection length of the local structure vector, $\|\tilde{V}_{\mathcal{S}}^{\mathrm{T}}a_i\|_2$ (as shown in the above second term), to compute the anomaly score. Notably, calculating this term is more efficient than calculating the reconstruction error, as it avoids projecting data back to their original space. Additionally, this term can also reflect the edge density of the corresponding local structure, making it a powerful indicator for structural anomalies that manifest as dense connections.

We summarize our proposed TFGAD in Algorithm 1. Noticeably, TFGAD offers several appealing advantages:

(1) **Training-free**: Since TFGAD's objective has a closed-form analytic solution, TFGAD requires no training parameters, reducing the risk of overfitting to anomalies while providing strong flexibility and generality in practice.

(2) **Computation-efficient**: The randomized SVD can solve TFGAD's objective within seconds with minimal memory overhead (no GPU required). The proposed lightweight scoring function further reduces computational overhead in detecting graph anomalies.

(3) **Scalable**: Due to computational efficiency, TFGAD possesses high scalability across various large-scale graphs with fast detection speed and limited memory requirements.

(4) **Easy-to-use**: The simple architecture of TFGAD enables an easy implementation with few lines of code (e.g., fewer than 10 lines in Python), facilitating its quick deployment across various applications. A PyTorch-like style pseudocode of TFGAD can be found in Appendix B.

---

**Algorithm 1** TFGAD: A training-free approach for graph anomaly detection

---

**Input:** Attribute matrix $X$, adjacency matrix $A$, hyper-parameters $k_{\mathcal{A}}$, $k_{\mathcal{S}}$, and $\eta$.
**Optimization Stage:**
Compute the approximated top $k_{\mathcal{A}}$ right singular vectors of $X$ via the randomized SVD, resulting in $\tilde{V}_{\mathcal{A}}$.
Compute the approximated top $k_{\mathcal{S}}$ right singular vectors of $A$ via the randomized SVD, resulting in $\tilde{V}_{\mathcal{S}}$.
**Test Stage:** Given a test node $v_i$, compute the anomaly score $\tau(v_i)$ via Equation (7).

---

## 4.4 Complexity Analysis

Let $\mathcal{G} = \{\mathcal{V}, X, A\}$ be an input graph with $m$ edges, where attribute matrix $X \in \mathbb{R}^{n \times d}$, adjacency matrix $A \in \mathbb{R}^{n \times n}$, and $m$ equals the number of non-zero elements in $A$. The complexity of randomized SVD on $X$ is $O(ndk_{\mathcal{A}})$, where $k_{\mathcal{A}}$ is the target rank for the decomposed matrices. Notably, the complexity of this operation on $A$ is $O(mk_{\mathcal{S}} + nk_{\mathcal{S}}^2)$, as $A$ can be efficiently implemented by a sparse

matrix, requiring solely the processing of its non-zero $m$ entries. For anomaly score computation, the complexity of calculating the reconstruction errors over $X$ is $O(2ndk_{\mathcal{A}})$, while the complexity of calculating the projection lengths over $A$ is $O(mk_{\mathcal{S}})$. Therefore, the overall complexity of TFGAD is $O(3ndk_{\mathcal{A}} + 2mk_{\mathcal{S}} + nk_{\mathcal{S}}^2)$, which is linearly dependent on the number of nodes and edges in the graph.

# 5 EXPERIMENTS

## 5.1 Experimental Setup

**Datasets.** Nine benchmark datasets are employed in our experiments: Cora, Citeseer, Pumbed, ACM, BlogCatalog (BCatalog for short), ogbn-Arxiv (Arxiv for short), ogbn-Products (Products for short), Books, and Reddit. Among these datasets, Cora, Citeseer, Pubmed, ACM, and BCatalog are five widely used small-scale benchmarks [42, 46, 47]. Arxiv and Products are two large-scale OGB datasets [19]. Since these datasets are free of anomalies, we follow the literature [5, 29] to inject synthetic anomalies. We refer readers to [27] for a detailed description of the standard injection approach. For a fair comparison with state-of-the-art baselines, we directly use the anomaly-injected small-scale datasets provided by [5] and follow its setup to inject large-scale datasets. Additionally, Books [41] and Reddits [22, 51], two datasets with real anomalies, are also employed for more comprehensive evaluation. The statistics of these datasets are summarized in Table 2. More details can be found in Appendix A.

**Table 2: Statistics of the datasets.**

| Dataset | # Nodes | # Edges | # Attributes | # Anomalies |
|---|---|---|---|---|
| Cora | 2,708 | 5,429 | 1,433 | 150 |
| Citeseer | 3,327 | 4,732 | 3,703 | 150 |
| Pubmed | 19,717 | 44,338 | 500 | 600 |
| ACM | 16,484 | 71,980 | 8,337 | 600 |
| BCatalog | 5,196 | 171,743 | 8,189 | 300 |
| Arxiv | 169,343 | 1,166,243 | 128 | 6000 |
| Products | 2,449,029 | 61,859,140 | 100 | 90000 |
| Books | 1,418 | 3,695 | 21 | 28 |
| Reddit | 10,984 | 168,016 | 64 | 366 |

**Baselines.** Eleven state-of-the-art baselines are utilized. DOMINANT (DOMT for short) [7], AnomalyDAE (ADAE for short) [13], and AdONE (AONE for short) [1] are reconstruction-based methods. CoLA (CLA for short) [29], ANEMONE (ANEM for short) [21], SL-GAD (SLGAD for short) [59], CONAD [54], and Sub-CR (SCR for short) [55] are contrastive-based methods. ResGCN (RGCN for short) [32], ComGA (CGA for short) [30], and GADAM [5] are methods with particularly designed message passing-oriented strategies.
**Evaluation Metrics.** Following the mainstream experimental setup of this research line [27, 48], two evaluation metrics are employed: the area under the Receiver-Operating-Characteristic curve (AUC-ROC) and the area under the Precision-Recall curve (AUC-PR). These two metrics evaluate the detection performance without posing any assumption on the anomaly threshold. AUC-ROC calculates the area under the ROC curve, which plots the true positive rate

**Table 3: Detection accuracy (AUC-ROC/AUC-PR) of TFGAD and its competing methods. The best accuracy per dataset is boldfaced, while the second-best is underlined. OOM indicates out-of-memory. The results are averages over five runs. Results for other methods are presented according to [5].**

| | Dataset | DOMT | ADAE | AONE | CLA | ANEM | SLGAD | CONAD | SCR | RGCN | CGA | GADAM | TFGAD |
|---|---|---|---|---|---|---|---|---|---|---|---|---|---|
| **AUC-ROC** | Cora | 0.8493 | 0.8431 | 0.8561 | 0.8801 | 0.9054 | 0.8983 | 0.7423 | 0.9132 | 0.8479 | 0.8840 | 0.9556 | **0.9867** |
| | Citeseer | 0.8391 | 0.8264 | 0.8724 | 0.8891 | 0.9239 | 0.9106 | 0.7145 | 0.9310 | 0.7647 | 0.9167 | 0.9415 | **0.9895** |
| | Pubmed | 0.8013 | 0.8973 | 0.7952 | 0.9535 | 0.9464 | 0.9476 | 0.6993 | 0.9629 | 0.8079 | 0.9212 | 0.9581 | **0.9828** |
| | ACM | 0.7452 | 0.7516 | 0.7219 | 0.7783 | 0.8802 | 0.8538 | 0.6849 | 0.7245 | 0.7681 | 0.8496 | 0.9603 | **0.9677** |
| | BCatalog | 0.7531 | 0.7658 | 0.7314 | 0.7807 | 0.8005 | 0.8037 | 0.6557 | 0.8071 | 0.7852 | 0.8030 | **0.8117** | 0.8042 |
| | Arxiv | OOM | OOM | OOM | OOM | OOM | OOM | OOM | OOM | OOM | OOM | 0.8122 | **0.9644** |
| | Products | OOM | OOM | OOM | OOM | OOM | OOM | OOM | OOM | OOM | OOM | **0.8499** | 0.7434 |
| | Books | 0.5012 | 0.5567 | 0.5366 | 0.3982 | 0.4341 | 0.5655 | 0.5224 | 0.5713 | 0.5665 | 0.5354 | 0.5983 | **0.7010** |
| | Reddit | 0.5621 | 0.5454 | 0.5015 | 0.5791 | 0.5563 | 0.5625 | 0.5610 | 0.5563 | 0.5012 | 0.5682 | 0.5809 | **0.6021** |
| | mean | 0.5613 | 0.5767 | 0.5572 | 0.5843 | 0.6052 | 0.6158 | 0.5089 | 0.6074 | 0.5602 | 0.6087 | 0.8298 | **0.8582** |
| | mean rank | 7.6667 | 7.5556 | 8.1111 | 5.8889 | 5.5556 | 4.4444 | 9.1111 | 4.1111 | 7.3333 | 5.1111 | 1.8889 | **1.3333** |
| **AUC-PR** | Cora | 0.2010 | 0.2831 | 0.2331 | 0.4700 | 0.4483 | 0.5232 | 0.2101 | 0.6240 | 0.4469 | 0.5799 | 0.7280 | **0.8197** |
| | Citeseer | 0.2106 | 0.2464 | 0.3065 | 0.3846 | 0.4211 | 0.4383 | 0.3065 | 0.4867 | 0.6446 | 0.5823 | 0.7512 | **0.8364** |
| | Pubmed | 0.3176 | 0.3037 | 0.3733 | 0.4350 | 0.4644 | 0.4861 | 0.4038 | 0.5413 | 0.3648 | 0.5247 | 0.4264 | **0.5830** |
| | ACM | 0.1774 | 0.2626 | 0.2638 | 0.3465 | 0.3399 | 0.3915 | 0.3612 | 0.4310 | 0.3804 | 0.4128 | **0.4446** | 0.4337 |
| | BCatalog | 0.1519 | 0.1658 | 0.1811 | 0.1964 | 0.1804 | 0.2683 | 0.2132 | 0.2438 | 0.2205 | 0.2579 | **0.2960** | 0.2750 |
| | Arxiv | OOM | OOM | OOM | OOM | OOM | OOM | OOM | OOM | OOM | OOM | 0.1948 | **0.2079** |
| | Products | OOM | OOM | OOM | OOM | OOM | OOM | OOM | OOM | OOM | OOM | 0.2469 | **0.2840** |
| | Books | 0.0190 | 0.0194 | 0.0202 | 0.0023 | 0.0072 | 0.0123 | 0.0192 | 0.0213 | 0.0179 | 0.0259 | 0.0279 | **0.0571** |
| | Reddit | 0.0370 | 0.0400 | 0.0320 | 0.0437 | 0.0415 | 0.0330 | 0.0326 | 0.0463 | 0.0396 | 0.0461 | **0.0481** | 0.0423 |
| | mean | 0.1238 | 0.1468 | 0.1567 | 0.2087 | 0.2114 | 0.2392 | 0.1718 | 0.2660 | 0.2350 | 0.2700 | 0.3515 | **0.3932** |
| | mean rank | 9.1111 | 8.1111 | 7.7778 | 6.4444 | 6.7778 | 5.4444 | 7.3333 | 3.3333 | 6.2222 | 3.4444 | 2.2222 | **1.6667** |

against the false positive rate at different thresholds. AUC-PR calculates the area under the PR curve, which plots precision against recall at different thresholds.

**Implementation Details.** The implementation of baselines is directly taken from the PyGOD package [26] if they are available; otherwise, from their provided source code. For TFGAD, its hyper-parameter search space is $k_{\mathcal{A}}$ in $\{1, 10\}$, $k_{\mathcal{S}}$ in $\{1, 5, 35, 60, 220, 600\}$, and $\eta$ in $\{0.05, 1, 10, 100, 200, 500\}$. Detailed hyper-parameter settings can be found in Appendix B. All experiments are conducted with an Intel AMD EPYC CPU with 12 cores, 60GB RAM, and a single NVIDIA RTX 4090 GPU with 25GB memory. The source code of TFGAD is available on GitHub.

## 5.2 Effectiveness Evaluation

Table 3 illustrates the detection performance in terms of AUC-ROC and AUC-PR of our proposed TFGAD and its competing methods. TFGAD achieves the highest mean performance over nine benchmark datasets on both metrics, where the mean rank of TFGAD is significantly higher than all baselines. TFGAD averagely obtains 4.5%-35.1% AUC-ROC improvements on eight out of nine datasets and 5.4%-129.3% AUC-PR gains on all nine datasets. Particularly, on the popular benchmark Cora, TFGAD raises the state-of-the-art AUC-PR by 9.17 points (from 0.7280 to 0.8197) and achieves the highest AUC-ROC of 0.9867. On other small-scale benchmarks (Citeseer and Pubmed), TFGAD exhibits consistent improvements in both AUC-ROC and AUC-PR. Impressively, TFGAD exhibits good

scalability on large-scale benchmarks, with a 15.22-point AUC-ROC from 0.8122 to 0.9644. Although TFGAD performs less effectively than GADAM on Products, it still obtains the best AUC-PR on that dataset. On benchmarks with real anomalies (Books and Reddit), TFGAD raises the state-of-the-art AUC-ROC by 10.27 points (from 0.5983 to 0.7010) on Books and doubles the AUC-PR (from 0.0279 to 0.0571). TFGAD also achieves the highest AUC-ROC of 0.6021 on Reddit. The above results demonstrate the superiority of TFGAD in detecting diverse anomalies with both synthetic and real patterns. Note that contrastive-based graph anomaly detection methods generally show more competitive performance than reconstruction-based counterparts. Nevertheless, TFGAD, built upon a minimalistic reconstruction-based framework, still outperforms the contrastive-based baselines, e.g., by a large margin in AUC-ROC on some challenging datasets like ACM and Arxiv. The superiority of TFGAD validates its effectiveness in accurately detecting graph anomalies with impressive scalability.

## 5.3 Efficiency Evaluation

Table 4 further illustrates the efficiency and scalability of TFGAD in terms of runtime (in seconds) and GPU overhead (in MB). To ensure a fair comparison with state-of-the-art baselines, runtime is evaluated across data processing, model training (when required), and anomaly score calculation, excluding the time spent on data loading and model initialization. GPU overhead, measured as peak

**Table 4: Efficiency Comparison of TFGAD and representative baselines in terms of runtime (in seconds) and GPU overhead (in MB). The best result per dataset is boldfaced, while the second-best is underlined. IMP indicates the improvement of TFGAD over the most efficient baseline: in runtime, the number of times TFGAD outperforms the baseline; in GPU overhead, the percentage improvement over the baseline.**

| | Method | Cora | Citeseer | Pubmed | ACM | BCatalog | Arxiv | Products | Books | Reddit |
|---|---|---|---|---|---|---|---|---|---|---|
| **Runtime (s)** | DOMT | 1.95 | 3.72 | 34.05 | 34.93 | 8.52 | OOM | OOM | 1.34 | 12.37 |
| | CLA | 164.56 | 358.89 | 1358.02 | 1110.32 | 330.91 | OOM | OOM | 57.36 | 484.37 |
| | GADAM | 1.25 | 1.41 | 7.60 | 3.64 | 8.83 | 318.19 | 770.63 | 0.68 | 1.83 |
| | TFGAD | **0.03** | **0.09** | **0.41** | **1.20** | **1.80** | **84.89** | **66.95** | **0.01** | **0.14** |
| | IMP | 41.67× | 15.67× | 18.54× | 3.03× | 4.73× | 3.75× | 11.51× | 68.00× | 13.07× |
| **GPU (MB)** | DOMT | 742 | 998 | 8,132 | 7,856 | 1,922 | OOM | OOM | 596 | 2,890 |
| | CLA | 624 | 696 | 2,100 | 2,686 | 1,024 | OOM | OOM | 558 | 1020 |
| | GADAM | 514 | 606 | 644 | 1,630 | 696 | 1,272 | 9,184 | 494 | 556 |
| | TFGAD | **0** | **0** | **0** | **0** | **0** | **0** | **0** | **0** | **0** |
| | IMP | 100% | 100% | 100% | 100% | 100% | 100% | 100% | 100% | 100% |

**Table 5: Comparison of TFGAD and its ablation variants in terms of AUC-ROC and runtime (in seconds). The best AUC-ROC per dataset is boldfaced, while the second-best is underlined.**

| | Method | Cora | Citeseer | Pubmed | ACM | BCatalog | Arxiv | Products | Books | Reddit |
|---|---|---|---|---|---|---|---|---|---|---|
| **AUC-ROC** | TFGAD$_\mathcal{A}$ | 0.7524 | 0.7286 | 0.7345 | 0.7438 | 0.7397 | 0.7319 | 0.7434 | 0.6772 | 0.5636 |
| | TFGAD$_\mathcal{S}$ | 0.7481 | 0.7907 | 0.7419 | 0.7535 | 0.6022 | 0.7530 | 0.5971 | 0.5580 | 0.5926 |
| | TFGAD$_\mathcal{R}$ | 0.9537 | 0.9606 | 0.9199 | 0.9024 | 0.7561 | OOM | OOM | 0.6700 | 0.5611 |
| | TFGAD$_\mathcal{P}$ | **0.9900** | **0.9933** | **0.9833** | **0.9784** | **0.8171** | 0.9521 | 0.7434 | 0.5000 | 0.5968 |
| | TFGAD | 0.9867 | 0.9895 | 0.9828 | 0.9677 | 0.8042 | **0.9644** | **0.7434** | **0.7010** | **0.6021** |
| **Runtime (s)** | TFGAD$_\mathcal{A}$ | 0.03 | 0.09 | 0.39 | 1.12 | 1.79 | 81.35 | 66.57 | 0.01 | 0.14 |
| | TFGAD$_\mathcal{S}$ | 0.02 | 0.05 | 0.38 | 0.83 | 1.70 | 82.13 | 65.15 | 0.01 | 0.14 |
| | TFGAD$_\mathcal{R}$ | 0.05 | 0.13 | 1.06 | 1.46 | 0.45 | OOM | OOM | 0.02 | 0.42 |
| | TFGAD$_\mathcal{P}$ | 0.10 | 0.61 | 1.41 | 5.52 | 2.52 | 84.53 | 82.02 | 0.01 | 0.13 |
| | TFGAD | 0.03 | 0.09 | 0.41 | 1.20 | 1.80 | 84.89 | 66.95 | 0.01 | 0.14 |

GPU consumption, is assessed throughout the entire process. Competing methods involve representatives in each baseline type, i.e., ADAE, CLA, and GADAM. those with the top two AUC-ROC performance in each type of baseline, i.e., DOMT, ADAE, CLA, SCR, and GADAM. The results are reported as averages over five runs. As shown in Table 4, TFGAD runs significantly faster than its competing methods, achieving speedups of 3.0× to 68.0× across various benchmarks. On large-scale benchmarks, TFGAD is 11.5× faster on Products and 3.75× faster on Arxiv compared to the most efficient baseline GADAM. Remarkably, TFGAD achieves superior efficiency without requiring GPU memory, leading to significantly better efficiency and scalability in practice, especially when GPU resources are limited. The dramatic boost in runtime and memory overhead of TFGAD validates its efficiency and scalability for graph anomaly detection.

## 5.4 Ablation Analysis

To further investigate the contribution of each component in TF-GAD, we compare TFGAD with four ablation variants: TFGAD$_\mathcal{A}$, TFGAD$_\mathcal{S}$, TFGAD$_\mathcal{R}$, and TFGAD$_\mathcal{P}$. TFGAD$_\mathcal{A}$ works solely with the reconstruction process of node attributes, while TFGAD$_\mathcal{S}$ employs only the projection process of local structures described in Section 4.3. TFGAD$_\mathcal{R}$ leverages the errors of reconstructing both node attributes and local structures for anomaly detection, aligning with the basic idea of reconstruction-based anomaly detection. Conversely, TFGAD$_\mathcal{P}$ leverages the projection lengths of node attributes and local structures.

Table 5 illustrates the comparison results of TFGAD and its ablation variants. TFGAD significantly outperforms TFGAD$_\mathcal{A}$ and TFGAD$_\mathcal{S}$ across all benchmarks while maintaining comparable runtime, demonstrating the effectiveness in incorporating the attribute reconstruction with the local structure projection for graph anomaly detection. Compared to TFGAD$_\mathcal{R}$, TFGAD achieves better performance in both accuracy and efficiency, which shows the superiority of projecting rather than reconstructing local structures for anomaly detection. Additionally, TFGAD achieves considerable performance on large-scale datasets, while TFGAD$_\mathcal{R}$ runs out of memory. This highlights the significant efficiency of projecting local structures compared to reconstructing them. Note that, on

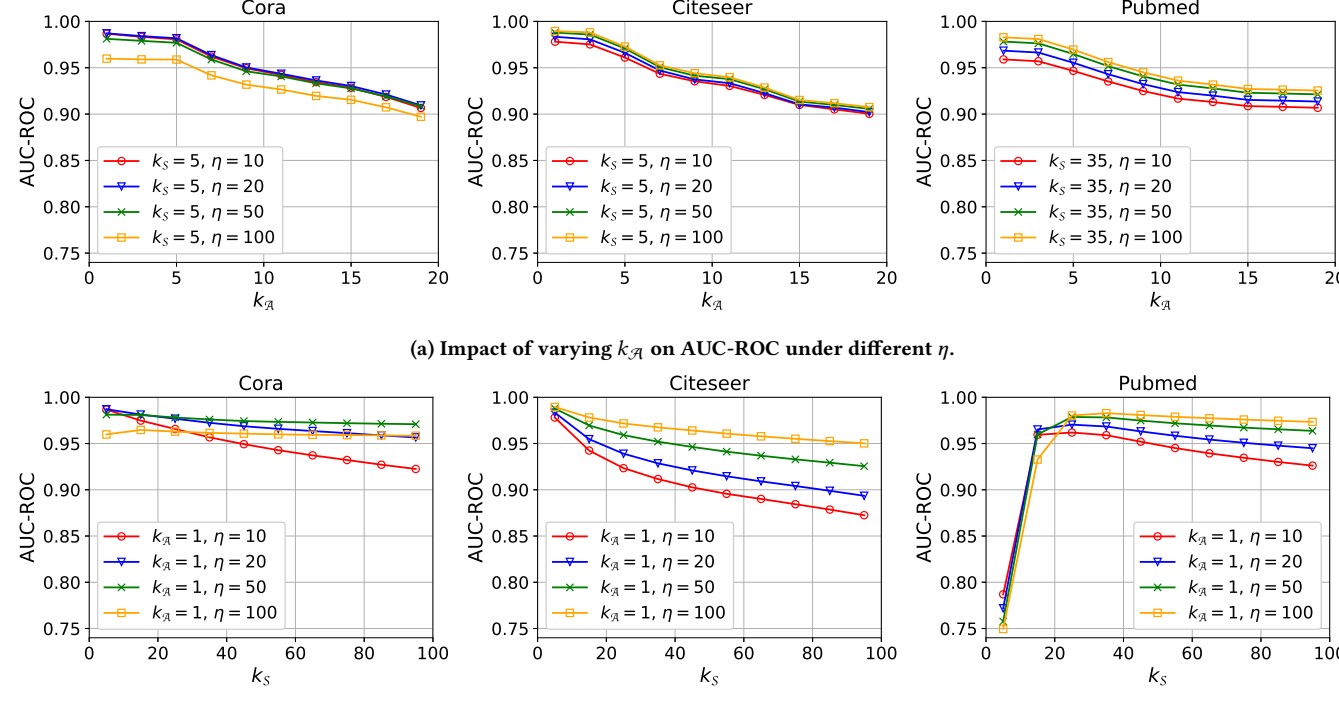

(a) Impact of varying $k_{\mathcal{A}}$ on AUC-ROC under different $\eta$.

(b) Impact of varying $k_{\mathcal{S}}$ on AUC-ROC under different $\eta$.

**Figure 4: AUC-ROC performance of TFGAD on Cora, Citeseer, and Pubmed w.r.t. hyper-parameters $k_{\mathcal{A}}$, $k_{\mathcal{S}}$, and $\eta$.**

small-scale benchmarks (Cora, Citeseer, Pubmed, ACM, and BCatalog), TFGAD performs relatively less effectively than TFGAD$_{\mathcal{P}}$, which suggests that the fully projection-based approach is more suitable in these cases. However, on large-scale benchmarks and those with real anomalies, TFGAD outperforms TFGAD$_{\mathcal{P}}$ by a significant margin, highlighting its flexibility and generality for detecting both synthetic and real anomalies.

## 5.5 Sensitivity Analysis

To investigate the impact of hyper-parameters $k_{\mathcal{A}}$, $k_{\mathcal{S}}$, and $\eta$ on performance, we conduct experiments on benchmark datasets: Cora, Citeseer, and Pubmed. Specifically, we evaluate the AUC-ROC performance of TFGAD by varying $k_{\mathcal{A}}$ from 1 to 19 with fixed $k_{\mathcal{S}}$ under different $\eta$ values (10, 20, 50, 100) and similarly varying $k_{\mathcal{S}}$ from 5 to 95 with fixed $k_{\mathcal{A}}$ under these $\eta$ values. As shown in Figure 4, we have the following observations: (1) The optimal hyper-parameters differ between datasets. For instance, the optimal $\eta$ for Citeseer and Pubmed is 100, but 10 for Cora. This implies that the contributions of attribute and structure patterns to anomaly detection are dataset-dependent. (2) Across different values of $\eta$, varying $k_{\mathcal{A}}$ shows similar trends in AUC-ROC performance, as does varying $k_{\mathcal{S}}$. This suggests that the choice of $\eta$ has minimal influence on the selection of $k_{\mathcal{A}}$ and $k_{\mathcal{S}}$. (3) After reaching an appropriate $\eta$, e.g., 10-100 for Cora and Citeseer, the AUC-ROC of TFGAD changes smoothly as the increasing of $k_{\mathcal{A}}/k_{\mathcal{S}}$, which demonstrates the weak sensitivity of TFGAD to these hyper-parameters.

## 6 CONCLUSION

In this paper, we propose TFGAD, a simple yet effective training-free approach for graph anomaly detection. TFGAD features compelling properties, including simplicity, efficiency, scalability, and ease of implementation. We start with a detailed review and analysis of issues in reconstruction-based anomaly detection methods, and then we discuss how they negatively impact detection performance. This analysis motivates a minimalistic, GNN-free, and modality-separate framework to detect graph anomalies. Based on this, TFGAD is built with minimum required linear transformations, each tailored to process a specific type of node feature (attributes or local structure). Remarkably, these transformations can be optimally determined via SVD techniques, thereby requiring no training parameters and GPU overhead. As a further improvement, the randomized SVD is employed to significantly reduce computational overhead. Additionally, a lightweight scoring function is adopted, which replaces the reconstruction of local structures by simply projecting them into a low-dimensional subspace, providing various advantages for detecting graph anomalies. Extensive experiments demonstrate substantial improvements of TFGAD over existing methods, where TFGAD achieves state-of-the-art performance in both accuracy (4.5%-35.1% AUC-ROC improvements and 5.4%-129.4% AUC-PR improvements) and time efficiency (3.0×-68.0× speedups) across various benchmark datasets.

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

## A   DATASET DETAILS

We provide additional information on the employed datasets, including the type of datasets, average degree, and the ratio of anomalies. The details are shown in Table 6.

**Table 6: Details of the datasets. The real-world datasets with Injected/Real (I/R) anomalies.**

| Dataset | I/R | # Nodes | # Edges | # Att. | Degree | # Ano. | Ratio |
|---------|-----|---------|---------|--------|--------|--------|-------|
| Cora | I | 2,708 | 5,429 | 1,433 | 2.0 | 150 | 5.5% |
| Citeseer | I | 3,327 | 4,732 | 3,703 | 1.4 | 150 | 4.5% |
| Pubmed | I | 19,717 | 44,338 | 500 | 2.3 | 600 | 3.1% |
| ACM | I | 16,484 | 71,980 | 8,337 | 4.4 | 600 | 3.6% |
| BCatalog | I | 5,196 | 171,743 | 8,189 | 33.1 | 300 | 5.7% |
| Arxiv | I | 169,343 | 1,166,243 | 128 | 6.9 | 6000 | 3.5% |
| Products | I | 2,449,029 | 61,859,140 | 100 | 25.3 | 90000 | 3.6% |
| Books | R | 1,418 | 3,695 | 21 | 2.6 | 28 | 2.0% |
| Reddit | R | 10,984 | 168,016 | 64 | 15.3 | 366 | 3.3% |

## B   DETAILED HYPER-PARAMETERS

Additional information about hyper-parameters of TFGAD are in Table 7.

**Table 7: Hyper-parameters of TFGAD for all used datasets.**

| | Cora | Citeseer | Pubmed | ACM | BCatalog | Arxiv | Products | Books | Reddit |
|---|------|----------|--------|-----|----------|-------|----------|-------|--------|
| $k_{\mathcal{A}}$ | 1 | 1 | 1 | 1 | 1 | 1 | 1 | 10 | 10 |
| $k_{\mathcal{S}}$ | 5 | 5 | 35 | 60 | 220 | 600 | 1 | 1 | 5 |
| $\eta$ | 10 | 100 | 100 | 10 | 0.05 | 1 | 10 | 200 | 500 |

## C   PSEUDOCODE OF TFGAD

Algorithm 2 provides the pseudocode of TFGAD in a PyTorch-like style, showing that TFGAD can be easily implemented with few lines of code.

**Algorithm 2** Pseudocode of TFGAD in a PyTorch-like style.

```
# att, adj: attribute and adjacency matrices
# k_att, k_adj: required numbers of top right singular vectors of
    attribute and adjacency matrices
# eta: balancing hyper-parameter

# perform randomized SVD
_, _, V_att = torch.svd_lowrank(att, q=k_att)
_, _, V_adj = torch.svd_lowrank(adj, q=k_adj)

# compute anomaly scores
att_rec_err = (att - att @ V_att @ V_att.T).pow(2).sum(dim=1)
adj_prj_len = (adj @ V_adj.pow(2).sum(dim=1)
y_score = att_rec_err + adj_prj_len / eta
```

## D   ANALYSIS OF ANOMALY DETECTION UNDER LIMITED DATA ACCESSIBILITY

We further analyze the performance of TFGAD under limited data accessibility, where only a subset of nodes is available for optimizing objective (5). We conduct experiments on popular benchmarks Cora, Citeseer, and Pubmed, using the default hyper-parameter settings presented in Table 7. Specifically, we evaluate the AUC-ROC performance of TFGAD across different ratios of available nodes for optimizing objective (5). For each ratio $r$ from 0.1 to 0.9, we randomly select $m = n * r$ nodes and utilize both their attributes and local structures to apply randomized SVD, as described in 4.2. The AUC-ROC is then measured on the entire input graph, considering

**Table 8: AUC-ROC performance of TFGAD under varying ratios of available nodes for optimizing objective (5).**

| | Dataset | 0.1 | 0.3 | 0.5 | 0.7 | 0.9 | 1.0 |
|---|---------|-----|-----|-----|-----|-----|-----|
| AUC-ROC | Cora | 0.9473 | 0.9567 | 0.9701 | 0.9811 | 0.9853 | 0.9867 |
| | Citeseer | 0.8936 | 0.9893 | 0.9865 | 0.9802 | 0.9874 | 0.9895 |
| | Pubmed | 0.8711 | 0.8779 | 0.9420 | 0.9583 | 0.9789 | 0.9828 |
| AUC-PR | Cora | 0.7263 | 0.6616 | 0.7245 | 0.7607 | 0.7997 | 0.8197 |
| | Citeseer | 0.6309 | 0.8548 | 0.8196 | 0.7783 | 0.8384 | 0.8364 |
| | Pubmed | 0.3381 | 0.3195 | 0.4609 | 0.5200 | 0.5443 | 0.5830 |

all nodes. The results (averaged over five runs) are presented in Table 8. Our key observations are as follows: (1) Despite limited access to input nodes (e.g., 30%-50% of nodes), TFGAD maintains competitive, and in some cases superior, performance compared to the best-performing baseline GADAM. For example, with only 30% of nodes accessible, TFGAD achieves the state-of-the-art AUC-ROC of 0.9567 on Cora and 0.9893 on Citeseer. (2) The AUC-ROC

performance of TFGAD improves as the ratio of accessible nodes increases, suggesting that TFGAD is less susceptible to anomalies within the available nodes. Although the AUC-PR of TFGAD fluctuates, it follows an overall upward trend and ultimately achieves state-of-the-art results. These results further demonstrate the superiority of TFGAD, highlighting its flexibility and generality in detecting graph anomalies across various scenarios.

