# OpenReview forum: "Training-free Graph Anomaly Detection: A Simple Approach via Singular Value Decomposition"
_ACM.org/TheWebConf/2025/Conference — WWW 2025 Poster_

### Official Review · Reviewer_LJPW · 2024-11-06

**Novelty:** 6
**Technical Quality:** 5

**Review:**

This paper presents a novel training-free approach for graph anomaly detection called TFGAD. The key highlights and evaluation of this work are listed below:
Strength
1. Originality and Significance: The paper identifies key issues in existing reconstruction-based graph anomaly detection methods and proposes a simple yet effective training-free approach to address them. Leveraging singular value decomposition (SVD), TFGAD can optimally determine the necessary transformation matrices without any training, making it a highly novel and significant contribution.
2. Performance: Extensive experiments in the paper demonstrate that TFGAD significantly outperforms state-of-the-art deep learning-based baselines across various benchmark datasets, with AUC-ROC improvements ranging from 4.5% to 35.1%. This showcases the effectiveness of the proposed approach.
3. Efficiency and Scalability: TFGAD achieves much lower runtime and memory overhead compared to the baselines, with speedups of 3.0x to 68.0x. This is a crucial advantage, as it makes TFGAD highly practical and scalable for large-scale graph data
4. Simplicity: The paper takes a minimalistic, GNN-free, and modality-separate approach, which is a significant departure from the complex deep learning-based methods. This simplicity makes TFGAD easy to implement and understand, further enhancing its practicality.
5. Comprehensive Evaluation: The authors conduct a thorough investigation of the issues in existing reconstruction-based methods and provide a detailed ablation study to validate the design choices of TFGAD. This comprehensive evaluation strengthens the rigor and credibility of the proposed approach.

Weakness

1. Theoretical Analysis: While the experimental results are compelling, the paper lacks a more in-depth theoretical analysis of the proposed method. A deeper understanding of the mathematical properties and guarantees of TFGAD would further strengthen the work.

2. Potential Limitations: The paper does not discuss potential limitations or failure cases of TFGAD, such as its sensitivity to certain types of graph structures or the impact of noise in the input data. Addressing these aspects could provide a more well-rounded understanding of the method.
3. Real-world Applications: The paper focuses on benchmark datasets and does not present any case studies or real-world use cases of TFGAD. Demonstrating the method's applicability and performance in practical scenarios would further enhance the significance of this work.
This paper presents a highly novel and impactful approach for graph anomaly detection. The simplicity, efficiency, and superior performance of TFGAD make it a significant contribution to the field of graph mining and security-related applications. With some additional theoretical analysis and exploration of real-world applications, this work has the potential to become a valuable reference in the research community.

**Questions:**

After reading this interesting research paper, we have some questions for the authors to discuss during the question period:

Theoretical Analysis and Guarantees

1. Can you provide more insights into the theoretical properties of the SVD-based transformations used in TFGAD?
2. What are the mathematical guarantees or bounds on the performance of your approach?
3. How does the choice of rank k in the truncated SVD impact the anomaly detection capabilities of TFGAD?
4. Can you provide guidelines or heuristics for selecting the optimal k?

Robustness and Limitations:

1. How robust is TFGAD to different types of graph anomalies, such as contextual anomalies, structural anomalies, or adversarial attacks?
2. Can you discuss any potential failure cases or limitations of your method?
3. What are the potential implications of noisy or incomplete input data (e.g., missing edges, corrupted attributes) on the performance of TFGAD?
4. How can the method be further enhanced to handle such real-world scenarios?

Real-world Applications and Deployment:

1. Can you share any insights or case studies on the application of TFGAD in real-world scenarios, such as social network analysis, financial fraud detection, or network security monitoring?
2. How does the method adapt to the challenges encountered in practice?
3. What are the key considerations for deploying TFGAD in large-scale, production-ready systems?
4. Can you discuss any engineering or system-level aspects that were not covered in the paper?

Comparison to Contrastive-based Methods:

1. The paper highlights the advantages of TFGAD over reconstruction-based methods, but how does it compare to the state-of-the-art contrastive-based approaches?
2. Can you provide more insights into the trade-offs between these two paradigms for graph anomaly detection?
3. Are there any potential synergies or ways to combine the strengths of TFGAD's training-free approach with the powerful representation learning capabilities of contrastive methods?

Future Research Directions:

1. What are the promising avenues for further extending or improving TFGAD?
2. Can you suggest any possible directions for future research building upon the foundations laid in this work?
3. Are there any other graph mining or security-related tasks where the SVD-based, training-free approach used in TFGAD could be applicable or beneficial?

**Reviewer Confidence:**

4: The reviewer is certain that the evaluation is correct and very familiar with the relevant literature

**Scope:**

4: The work is relevant to the Web and to the track, and is of broad interest to the community

---

### Official Review · Reviewer_5s9R · 2024-11-29

**Novelty:** 5
**Technical Quality:** 5

**Review:**

Summary:

This paper proposes a training-free graph anomaly detection method called TFGAD. It computes the transformation matrices for node attributes and local structures separately using Singular Value Decomposition (SVD), avoiding three issues associated with using GNNs. Additionally, a lightweight anomaly scoring function is employed. Through comprehensive comparative experiments and ablation studies, demonstrating that TFGAD can analyze graph anomaly detection tasks with lower computational overhead and time, while achieving higher accuracy.

Strengths:
1. The method proposed in this paper is interesting. By using SVD instead of GNNs, it reduces runtime and memory overhead, making it well-suited for large-scale datasets. Moreover, the time complexity of the method is analyzed, which helps to evaluate its applicability in different scenarios.

2. The experiments are thorough, covering 11 state-of-the-art baselines and 9 benchmark datasets, including two large-scale datasets.

3. The method and the figures are simple and intuitive, making it easy to understand and implement.

Weakness:
1. TFGAD differs from reconstruction-based graph anomaly detection in two aspects: whether SVD is used and whether the mapping matrices for node attributes and local structures are solved separately. The paper does not prove that both aspects work together to ensure that TFGAD outperforms reconstruction-based methods in graph anomaly detection.

2. The theoretical foundation for how SVD addresses the three issues mentioned in Section 4.1 is not provided, nor is there an analysis of the accuracy loss due to the use of random SVD.

**Questions:**

1. Does the pseudocode lack the code for solving the projection matrix?

2. The "x" in numbers like 2.3.0x and 68.0x slightly affects the reading experience.

3. Can other dimensionality reduction methods be used instead of SVD when solving the projection matrix? If so, how do their effects compare to those of SVD?

4. The analysis of limitations in existing methods in the abstract is somewhat vague. While the limitations become clear after reading the paper, is there a better way to express them? For instance, could the key limitations be highlighted or summarized with specific terms?

**Reviewer Confidence:**

2: The reviewer is willing to defend the evaluation, but it is likely that the reviewer did not understand parts of the paper

**Scope:**

3: The work is somewhat relevant to the Web and to the track, and is of narrow interest to a sub-community

---

### Official Review · Reviewer_cDmb · 2024-11-29

**Novelty:** 6
**Technical Quality:** 5

**Review:**

Summary:
This paper focuses on graph anomaly detection which is important in many web applications. This paper proposes TFGAD, a graph anomaly detection approach using the singular value decomposition (SVD) method for encoding and decoding the graph data to identify anomalies with large decoding/reconstruction errors. Experiments validate the effectiveness and efficiency of TFGAD.

Strengths:
1. Innovative and simple but effective method. Prior works based on learning (e.g., GNNs) suffer from a huge computation burden, while this method requires no training and solves the objective function with a closed-form analytic solution.
2. Scalability. This method can be deployed on graphs with millions of edges, which reflects the adaptability of this method to real-life scenarios.
3. Inspiring. Observational experiments in Section 4.1 provide some inspiring insights.
4. Clarity. The presentation is easy-to-follow and the problem statement is clear.

Weaknesses:
1. Clarity. The presentation of related works in Section 2 is not well-structured. E.g., In Section 2.1, the limitation analysis of prior works is inconsistent with that in Section 1, where reconstruction-based methods have no separate evaluation. The scalability issues seem to be the problem of both reconstruction- and contrastive-based methods in Section 2 while only to be a drawback of reconstruction-based methods in Section 1.
2. Quality. The three main insights in Section 4.1, which instructs the design of TFGAD are mainly derived from experimental ablation studies. I suggest adding more discussion or theoretical proof here. Also, the second term of the RHS of Equation 7 is replaced by a computationally efficient term, where the paper could be strengthened by deriving more formulas here to clarify the capability of this substitute.
3. Clarity. The introduction of baseline methods in experiments is too brief. More details are recommended to be supplemented in the appendix.

**Questions:**

Prior works (e.g., reference[28] in Section 2.1) using matrix factorization for graph anomaly detection are not explained in detail, how do these methods work? Can you provide more explanation or experimental results on them?

**Reviewer Confidence:**

3: The reviewer is confident but not certain that the evaluation is correct

**Scope:**

4: The work is relevant to the Web and to the track, and is of broad interest to the community

---

### Official Review · Reviewer_n6g2 · 2024-12-01

**Novelty:** 4
**Technical Quality:** 3

**Review:**

This paper proposes a simple effective training-free approach for graph anomaly detection which combats poor detection accuracy, long training time, complicated training schemes, and lack of scalability. The authors' perspective is novel and well researched for the problem of graph anomaly detection with less amount of data. However, there exists some issues to be resolved.

**Questions:**

1、In order to solve the problem of singular value decomposition matrix processing when the amount of data is very large, the author first inputs a small matrix. How is the small matrix selected and on what basis was it chosen？
2、The method proposed in this thesis does reduce the runtime and memory overhead on datasets with small data volumes, but how does the method perform for very large data volumes?

**Reviewer Confidence:**

4: The reviewer is certain that the evaluation is correct and very familiar with the relevant literature

**Scope:**

4: The work is relevant to the Web and to the track, and is of broad interest to the community

---

### Official Review · Reviewer_F2RN · 2024-12-03

**Novelty:** 3
**Technical Quality:** 3

**Review:**

This paper proposes a training-free graph anomaly detection method called TFGAD based on singular value decomposition.

**Pros:**
1. Extensive experiments conducted on 9 datasets to demonstrate effectiveness.
2. The model uses randomized SVD to make it efficient

**Cons:**
1. The novelty seems limited. Separately considering attributes and structure for anomaly detection is a common approach. It is also standard to use matrix decomposition [1], and this paper directly uses the existing randomized SVD.
2. The authors state that early shallow methods, such as matrix decomposition and residual analysis, struggle with complex graph information. However, SVD is essentially a matrix decomposition method and is typically computationally inefficient. A detailed discussion on why SVD performs well in this work and how it compares to other shallow techniques, particularly matrix decomposition methods like CUR, is necessary.
3. There are dimensional inconsistencies in key equations (e.g., Eq. 5 and 7). In Eq.5, A is [n, n], while WsWs^T is [d, d]. It is unclear how the operation AWsWs^T is performed.
4. In the effectiveness evaluation, deep learning methods run on GPUs, while SVD relies heavily on CPU and memory. Comparing only GPU memory consumption is unfair.
5. In Figure 4b, the trend of Ks on Pubmed differs from that on other datasets. Further analysis is needed to explain this discrepancy.
[1] Lee, Baozhen, et al. "Enhanced multi-view anomaly detection on attribute networks by truncated singular value decomposition." International Journal of Machine Learning and Cybernetics (2024): 1-19.

**Questions:**

See in Cons

**Reviewer Confidence:**

4: The reviewer is certain that the evaluation is correct and very familiar with the relevant literature

**Scope:**

3: The work is somewhat relevant to the Web and to the track, and is of narrow interest to a sub-community